# Compositional generalization through meta sequence-to-sequence learning

**Brenden M. Lake**
New York University
Facebook AI Reasearch
brenden@nyu.edu

## Abstract

People can learn a new concept and use it compositionally, understanding how to "blicket twice" after learning how to "blicket." In contrast, powerful sequence-to-sequence (seq2seq) neural networks fail such tests of compositionality, especially when composing new concepts together with existing concepts. In this paper, I show how memory-augmented neural networks can be trained to generalize compositionally through meta seq2seq learning. In this approach, models train on a series of seq2seq problems to acquire the compositional skills needed to solve new seq2seq problems. Meta se2seq learning solves several of the SCAN tests for compositional learning and can learn to apply implicit rules to variables.

## 1 Introduction

People can learn new words and use them immediately in a rich variety of ways, thanks to their skills in compositional learning. Once a person learns the meaning of the verb "to Facebook", she or he can understand how to "Facebook slowly," "Facebook eagerly," or "Facebook while walking." These abilities are due to systematic compositionality, or the algebraic capacity to understand and produce novel utterances by combining familiar primitives [5, 27]. The "Facebook slowly" example depends on knowledge of English, yet the ability is more general; people can also generalize compositionally when learning to follow instructions in artificial languages [18]. Despite its importance, systematic compositionality has unclear origins, including the extent to which it is inbuilt versus learned. A key challenge for cognitive science and artificial intelligence is to understand the computational underpinnings of human compositional learning and to build machines with similar capabilities.

Neural networks have long been criticized for lacking compositionality, leading critics to argue they are inappropriate for modeling language and thought [8, 24, 25]. Nonetheless neural architectures have advanced and made important contributions in natural language processing (NLP) [20]. Recent work has revisited these classic critiques through studies of modern neural architectures [10, 16, 3, 21, 23, 2, 6], with a focus on the sequence-to-sequence (seq2seq) models used successfully in machine translation and other NLP tasks [34, 4, 38]. These studies show that powerful seq2seq approaches still have substantial difficulties with compositional generalization, especially when combining a new concept ("to Facebook") with previous concepts ("slowly" or "eagerly") [16, 3, 21].

New benchmarks have been proposed to encourage progress [10, 16, 2], including the SCAN dataset for compositional learning [16]. SCAN involves learning to follow instructions such as "walk twice and look right" by performing a sequence of appropriate output actions; in this case, the correct response is to "WALK WALK RTURN LOOK." A range of SCAN examples are shown in Table 1. Seq2seq models are trained on thousands of instructions built compositionally from primitives ("look", "walk", "run", "jump", etc.), modifiers ("twice", "around right," etc.) and conjunctions ("and" and "after"). After training, the aim is to execute, zero-shot, novel instructions such as "walk around right after look twice." Previous studies show that seq2seq recurrent neural networks

(RNN) generalize well when the training and test sets are similar, but fail catastrophically when generalization requires systematic compositionality [16, 3, 21]. For instance, models often fail to understand how to "jump twice" after learning how to "run twice," "walk twice," and how to "jump." Building neural architectures with these compositional abilities remains an open problem.

In this paper, I show how memory-augmented neural networks can be trained to generalize compositionally through "meta sequence-to-sequence learning" (meta seq2seq learning). As is standard with meta learning, training is distributed across a series of small datasets called "episodes" instead of a single static dataset [36, 32, 7], in a process called "meta-training." Specific to meta seq2seq learning, each episode is a novel seq2seq problem that provides "support" sequence pairs (input and output) and "query" sequences (input only), as shown in Figures 1 and 2. The network loads the support sequence pairs into an external memory [33, 11, 31] to provide needed context for producing the right output sequence for each query sequence. The network's output sequences are compared to the targets, demonstrating how to generalize compositionally from the support items to the query items.

Meta seq2seq networks meta-train on multiple seq2seq problems that require compositional generalization, with the aim of acquiring the compositional skills needed to solve new problems. New seq2seq problems are solved entirely using the activation dynamics and external memory of the networks; no weight updates are made after the meta-training phase ceases. Through its unique choice of architecture and training procedure, the network can implicitly learn rules that operate on variables, an ability considered beyond the reach of eliminative connectionist networks [24, 25, 23] but which has been pursued by more structured alternatives [33, 11, 12, 28]. In the sections below, I show how meta seq2seq learning can solve several challenging SCAN tasks for compositional learning, although generalizing to longer output sequences remains unsolved.

## 2   Related work

Meta sequence-to-sequence learning builds on several areas of active research. Meta learning has been successfully applied to few-shot image classification [36, 32, 7, 19] including sequential versions that require external memory [31]. Few-shot visual tasks are qualitatively different from the compositional reasoning tasks studied here, which demand different architectures and learning principles. Closer to the present work, meta learning has been recently applied to low resource machine translation [13], demonstrating one application of meta learning to seq2seq translation problems. Crucially, these networks tackle a new task through weight updates rather than through memory and reasoning [7], and it is unclear whether this approach would work for compositional reasoning.

External memories have also expanded the capabilities of modern neural network architectures. Memory networks have been applied to reasoning and question answering tasks [33], in cases where only a single output is needed instead of a series of outputs. The Differentiable Neural Computer (DNC) [11] is also related to my proposal, in that a single architecture can reason through a wide range of scenarios, including seq2seq-like graph traversal tasks. The DNC is a complex architecture with multiple heads for reading and writing to memory, temporal links between memory cells, and trackers to monitor memory usage. In contrast, the meta seq2seq learner uses a simple memory mechanism akin to memory networks [33] and does not call the memory module with every new input symbol. Meta seq2seq uses higher-level abstractions to store and reason with entire sequences.

There has been recent progress on SCAN due to clever data augmentation [1] and syntax-based attention [30], although both approaches are currently limited in scope. For instance, syntactic attention relies on a symbol-to-symbol mapping module that may be inappropriate for many domains. Meta seq2seq is compared against syntactic attention [30] in the experiments that follow.

## 3   Model

The meta sequence-to-sequence approach *learns how to learn* sequence-to-sequence (seq2seq) problems – it uses a series of training seq2seq problems to develop the needed compositional skills for solving new seq2seq problems. An overview of the meta seq2seq learner is illustrated in Figure 1. In this figure, the network is processing a query instruction "jump twice" in the context of a support set that shows how to "run twice," "walk twice", "look twice," and "jump." In broad strokes, the architecture is a standard seq2seq model [22] translating a query input into a query output (Figure 1). A recurrent neural network (RNN) encoder ($f_{ie}$; red RNN in bottom right of Figure 1) and a RNN

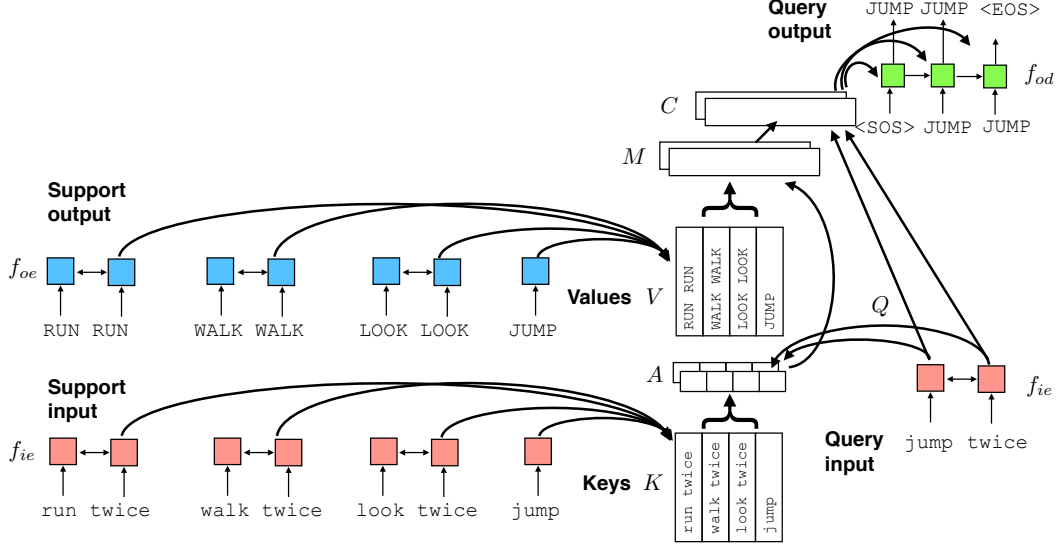

Figure 1: The meta sequence-to-sequence learner. The backbone is a sequence-to-sequence (seq2seq) network augmented with a context $C$ produced by an external memory. The seq2seq model uses an RNN encoder ($f_{ie}$; bottom right) to read a query and then pass stepwise messages $Q$ to an attention-based RNN decoder ($f_{od}$; top right). Distinctive to meta seq2seq learning, the messages $Q$ are transformed into $C$ based on context from the support set (left). The transformation operates through a key-value memory. Support item inputs are encoded and used a keys $K$ while outputs are encoded and used as value $V$. The query is stepwise compared to the keys, retrieving weighted sums $M$ of the most similar values. This is mapped to $C$ which is decoded as the final output sequence. Color coding indicates shared RNN modules.

decoder ($f_{od}$; green RNN in top right of Figure 1) work together to interpret the query sequence as an output sequence, with the encoder passing an embedding at each timestep ($Q$) to a Luong attention decoder [22]. The architecture differs from standard seq2seq modeling through its use of the support set, external memory, and training procedure. As the messages pass from the query encoder to the query decoder, they are infused with stepwise context $C$ provided by an external memory that stores the support items. The inner-working of the architecture are described in detail below.

**Input encoder.** The input encoder $f_{ie}(\cdot, \cdot)$ (Figure 1 red) encodes the query input instruction (e.g., "jump twice") and each of the input instructions for the $n_s$ support items ("run twice", "walk twice", "jump", etc.). The encoder first embeds the sequence of symbols (e.g., words) to get a sequence of input embeddings $w_t \in \mathbb{R}^m$, which the RNN transforms into hidden embeddings $h_t \in \mathbb{R}^m$,

$$h_t = f_{ie}(h_{t-1}, w_t). \tag{1}$$

For the query sequence, the embedding $h_t$ at each step $t = 1, \ldots, T$ passes both through the external memory as well as directly to the decoder. For each support sequence, only the last step hidden embedding is needed, denoted $K_i \in \mathbb{R}^m$ for $i = 1, \ldots, n_s$. These vectors $K_i$ become the keys in the external key-value memory (Figure 1). Although other choices are possible, this paper uses bidirectional long short-term memory encoders (biLSTM) [14].

**Output encoder.** The output encoder $f_{oe}(\cdot, \cdot)$ (Figure 1 blue) is used for each of the of $n_s$ support items and their output sequences (e.g., "RUN RUN", "WALK WALK", "JUMP", etc.). First, the encoder embeds the sequence of output symbols (e.g., actions) using an embedding layer. Second, a single embedding for the entire sequence is computed using the same process as $f_{ie}(\cdot, \cdot)$ (Equation 1). Only the final RNN state is captured for each support item $i$ and stored as the value vector $V_i \in \mathbb{R}^m$ for $i = 1, \ldots, n_s$ in the key-value memory. A biLSTM encoder is also used.

**External memory.** The architecture uses a soft key-value memory similar to memory networks [33]. The precise formulation used is described in [35]. The key-value memory uses the attention function

$$\text{Attention}(Q, K, V) = \text{softmax}(\frac{QK^T}{\sqrt{m}})V = AV, \tag{2}$$

with matrices $Q$, $K$, and $V$ for the queries, keys, and values respectively, and the matrix $A$ as the attention weights, $A = \text{softmax}(QK^T/\sqrt{m})$. Each query instruction spawns $T$ embeddings

from the RNN encoder, one for each query symbol, which populate the rows of the query matrix $Q \in \mathbb{R}^{T,m}$. The encoded support items form the rows of $K \in \mathbb{R}^{n_s,m}$ and the rows of $V \in \mathbb{R}^{n_s,m}$ for their input and output sequences, respectively. Attention weights $A \in \mathbb{R}^{T,n_s}$ indicate which memory cells are active for each query step. The output of the memory is a matrix $M = AV$ where each row is a weighted combination of the value vectors, indicating the memory output for each of the $T$ query input steps, $M \in \mathbb{R}^{T,m}$. Finally, a stepwise context is computed by combining the query input embeddings $h_t$ and the stepwise memory outputs $M_t \in \mathbb{R}^m$ with a concatenation layer $C_t = \tanh(W_{c_1}[h_t; M_t])$ producing a stepwise context matrix $C \in \mathbb{R}^{T,m}$.

For additional representational power, the key-value memory could replace the simple attention module with a multi-head attention module, or even a transformer-style multi-layer multi-head attention module [35]. This additional power was not needed for the tasks tackled in this paper, but it is compatible with the meta seq2seq approach.

**Output decoder.** The output decoder translates the stepwise context $C$ into an output sequence (Figure 1 green). The decoder embeds the previous output symbol as vector $o_{j-1} \in \mathbb{R}^m$ which is fed to the RNN (LSTM) along with the previous hidden state $g_{j-1} \in \mathbb{R}^m$ to get the next hidden state,

$$g_j = f_{od}(g_{j-1}, o_{j-1}). \tag{3}$$

The initial hidden state $g_0$ is set as the context from the last step $C_T \in \mathbb{R}^m$. Luong-style attention [22] is used to compute a decoder context $u_j \in \mathbb{R}^m$ such that $u_j = \text{Attention}(g_j, C, C)$. This context is passed through another concatenation layer $\widetilde{g}_j = \tanh(W_{c_2}[g_j; u_j])$ which is then mapped to a softmax output layer to produce an output symbol. This process repeats until all of the output symbols are produced and the RNN terminates the response by producing an end-of-sequence symbol.

**Meta-training.** Meta-training optimizes the network across a series of training episodes, each of which is a novel seq2seq problem with $n_s$ support items and $n_q$ query items (see example in Figure 2). The model's vocabulary is the union of the episode vocabularies, and the loss function is the negative log-likelihood of the predicted output sequences for the queries. My implementation uses each episode as a training batch and takes one gradient step per episode. For improved sample and training efficiency, the optimizer could take multiple steps per episode or replay past episodes, although this was not explored here.

During meta-training, the network may need extra encouragement to use its memory. To provide this, the support items are passed through the network as additional query items, i.e. using an auxiliary "support loss" that is added to the query loss computed from the query items. The support items have already been observed and stored in memory, and thus it is not noteworthy that the network learns to reconstruct these output sequences. Nevertheless, it amplifies the memory during meta-training.

## 4 Experiments

### 4.1 Architecture and training parameters

A PyTorch implementation is available (see acknowledgements). All experiments use the same hyperparameters, and many were set according to the best-performing seq2seq model in [16]. The input and output sequence encoders are two-layer biLSTMs with $m = 200$ hidden units per layer, producing $m$ dimensional embeddings. The output decoder is a two-layer LSTM also with $m = 200$. Dropout is applied with probability 0.5 to each LSTM and symbol embedding. A greedy decoder is effective due to SCAN's determinism [16].

Networks are meta-trained for 10,000 episodes with the ADAM optimizer [15]. The learning rate is reduced from 0.001 to 0.0001 halfway, and gradients with a $l_2$-norm greater than 50 are clipped. With my PyTorch implementation, it takes less than 1 hour to train meta seq2seq on SCAN using one NVIDIA Titan X GPU (regular seq2seq trains in less than 30 minutes). All models were trained five times with different random initializations and random meta-training episodes.

### 4.2 Experiment: Mutual exclusivity

This experiment evaluates meta seq2seq learning on a synthetic task borrowed from developmental psychology (Figure 2). Each episode introduces a new mapping from non-sense words ("dax", "wif", etc.) to non-sense meanings ("red circle", "green circle", etc.), partially revealed in the support set.

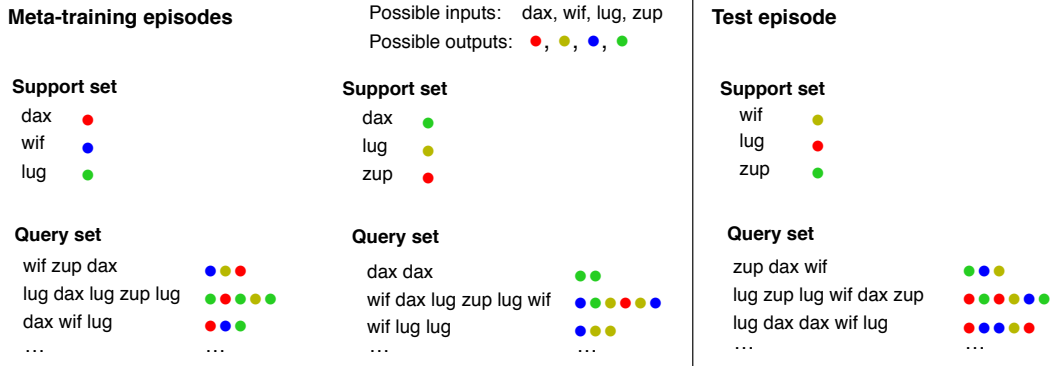

Figure 2: The mutual exclusivity task showing two meta-training episodes (left) and one test episode (right). Each episode requires executing instructions in a novel language of 4 input pseudowords ("dax", "wif", etc.) and four output actions ("red", "yellow", etc.). Each episode has a random mapping from pseudowords to meanings, providing three isolated words and their outputs as support. Answering queries requires concatenation as well as reasoning by mutual exclusivity to infer the fourth mapping ("dax" means "blue" in the test episode).

To answer the queries, a model must acquire two abilities inspired by human generalization patterns [18]: 1) using isolated symbol mappings to translate concatenated symbol sequences, and 2) using mutual exclusivity (ME) to resolve unseen mappings. Children use ME to help learn the meaning of new words, assuming that an object with one label does not need another [26]. When provided with a familiar object (e.g., a cup) and an unfamiliar object (e.g., a cherry pitter) and asked to "Show me the dax," children tend to pick the unfamiliar object rather than the familiar one.

Adults also use ME to help resolve ambiguity. When presented with episodes like Figure 2 in a laboratory setting, participants use ME to resolve unseen mappings and translate sequences in a symbol-by-symbol manner. Most people generalize in this way *spontaneously*, without any instructions or feedback about how to respond to compositional queries [18]. An untrained meta seq2seq learner would not be expected to generalize spontaneously – human participants come to the task with a starting point that is richer in every way – but computational models should nonetheless be capable of these inferences if trained to make them. This is a challenge for neural networks because the mappings change every episode, and standard architectures do not reason using ME. In fact, standard networks map novel inputs to familiar outputs, which is the opposite of ME [9].

**Experimental setup.** During meta-training, each episode is generated by sampling a random mapping from four input symbols to four output symbols (19 permutations used for meta-training and 5 for testing). The support set shows how three symbols should be translated, while one is withheld. The queries consist of arbitrary concatenations of the pseudowords (length 2 to 6) which can be translated symbol-by-symbol to produce the proper output responses (20 queries per episode). The fourth input symbol, which was withheld from the support, is used in the queries. The model must learn how to use ME to map this unseen symbol to an unseen meaning rather than a seen meaning (Figure 2).

**Results.** Meta seq2seq successfully learns to reason with ME to answer queries, achieving 100% accuracy (SD = 0%). Based on the isolated mappings stored in memory, the network learns to translate sequences of those items. Moreover, it can acquire and use new mappings at test time, utilizing only its external memory and the activation dynamics. By learning to use ME, the network shows it can reason about the *absence* of symbols in the memory rather than simply their presence. The attention weights and use of memory is visualized and presented in the appendix (Figure A.1).

### 4.3 Experiment: Adding a new primitive through permutation meta-training

This experiment evaluates meta seq2seq learning on the SCAN task of adding a new primitive [16]. Models are trained to generalize compositionally by decomposing the original SCAN task into a series of related seq2seq sub-tasks. The goal is to learn a new primitive instruction and use it compositionally, operationalized in SCAN as the "add jump" split [16]. Models learn a new primitive "jump" and aim to use it in combination with other instructions, resembling the "to Facebook" example introduced earlier in this paper. First, the original seq2seq problem from [16] is described. Second, the adapted problem for training meta seq2seq learners is described.

Table 1: SCAN task for compositional learning with input instructions (left) and their output actions (right) [16].

| | | |
|---|---|---|
| jump | ⇒ | JUMP |
| jump left | ⇒ | LTURN JUMP |
| jump around right | ⇒ | RTURN JUMP RTURN JUMP RTURN JUMP RTURN JUMP |
| turn left twice | ⇒ | LTURN LTURN |
| jump thrice | ⇒ | JUMP JUMP JUMP |
| jump opposite left and walk thrice | ⇒ | LTURN LTURN JUMP WALK WALK WALK |
| jump opposite left after walk around left | ⇒ | LTURN WALK LTURN WALK LTURN WALK LTURN WALK LTURN LTURN JUMP |

**Seq2seq learning.** Standard seq2seq models applied to SCAN have both a training and a test phase. During training, seq2seq models are exposed to the "jump" instruction in a single context demonstrating how to jump in isolation. Also during training, the models are exposed to all primitive and composed instructions for the other actions (e.g., "walk", "walk twice", "look around right and walk twice", etc.) along with the correct output sequences, which is about 13,000 unique instructions. Following [16], the critical "jump" demonstration is overrepresented in training to ensure it is learned.

During test, models are evaluated on all of the composed instructions that use the "jump" primitive, examining the ability to integrate new primitives and use them productively. For instance, models are evaluated on instructions such as "jump twice", "jump around right and walk twice", "walk left thrice and jump right thrice," along with about 7,000 other instructions using jump.

**Meta seq2seq learning.** Meta seq2seq models applied to SCAN have both a meta-training and a test phase. During meta-training, the models observe episodes that are variants of the original seq2seq problem, each of which requires rapid learning of new meanings for the primitives. Specifically, each meta-training episode provides a different random assignment of the primitive instructions ('jump','run', 'walk', 'look') to their meanings ('JUMP','RUN','WALK','LOOK'), with the restriction that the proper (original) permutation not be observed during meta-training. Withholding the original permutation, there are 23 possible permutations for meta-training. Each episode presents 20 support and 20 query instructions, with instructions sampled from the full SCAN set. The models predict the response to the query instructions, using the support instructions and their outputs as context. Through meta-training, the models are familiarized with all of the possible SCAN training and test instructions, but no episode maps all of its instructions to their original (target) outputs sequences. In fact, models have no signal to learn which primitives in general correspond to which actions, since the assignments are sampled anew for each episode.

During test, models are evaluated on rapid learning of new meanings. Just four support items are observed and loaded into memory, consisting of the isolated primitives ('jump','run', 'walk', 'look') paired with their original meanings ('JUMP','RUN','WALK','LOOK'). Notably, memory use at test time (with only four primitive items in memory) diverges substantially from memory use during meta-training (with 20 complex instructions in memory). To evaluate test accuracy, models make predictions on the original SCAN test instructions consisting of all composed instructions using "jump." An output sequence is considered correct only if it perfectly matches the target sequence.

**Alternative models.** The meta seq2seq learner is compared with an analogous "standard seq2seq" learner [22], which uses the same architecture with the external memory removed. The standard seq2seq learner is trained on the original SCAN problem with a fixed meaning for each primitive. Each meta seq2seq "episode" can be interpreted as a standard seq2seq "batch," and a batch size of 40 is chosen to equate the total number of presentations between approaches. All other architectural and training parameters are shared between meta seq2seq learning and seq2seq learning.

The meta seq2seq learner is also compared with two additional lesioned variants that examine the importance of different architectural components. First, the meta seq2seq learner is trained "without support loss" (Section 3 meta-training), which guides the architecture about how to best use its memory. Second, the meta seq2seq learner is trained "without decoder attention" (Section 3 output decoder). This leads to substantial differences in the architecture operation; rather than producing a sequence of context embeddings $C_1, \ldots, C_T$ for each step of the $T$ steps of a query sequence, only the last step context $C_T$ is computed and passed to the decoder.

**Results.** The results are summarized in Table 2. On the "add jump" test set [16], standard seq2seq modeling completely fails to generalize compositionally, reaching an average performance of only 0.03% correct (SD = 0.02). It fails even while achieving near perfect performance on the training set

Table 2: Test accuracy on the SCAN "add jump" task across different training paradigms.

| Model | standard training | permutation meta-training | augmentation meta-training |
|---|---|---|---|
| meta seq2seq learning | — | **99.95%** | **98.71%** |
| -without support loss | — | 5.43% | **99.48%** |
| -without decoder attention | — | 10.32% | 9.29% |
| standard seq2seq | 0.03% | — | 12.26% |
| syntactic attention [30] | 78.4% | — | — |

(>99% on average). This replicates the results from [16] which trained many seq2seq models, finding the best network performed at only 1.2% accuracy. Again, standard seq2seq models do not show the necessary systematic compositionality.

The meta seq2seq model succeeds at learning compositional skills, achieving an average performance of 99.95% correct (SD = 0.08). At test, the support set contains only the four primitives and their mappings, demonstrating that meta seq2seq learning can handle test episodes that are qualitatively different from those seen during training. Moreover, the network learns how to store and retrieve variables from memory with arbitrary assignments, as long as the network is familiarized with the possible input and output symbols during meta-training (but not necessarily how they correspond). A visualization of how meta seq2seq uses attention on SCAN is shown in the appendix (Figure A.2). The meta seq2seq learner also outperforms syntactic attention which achieves 78.4% and varies widely in performance across runs (SD = 27.4) [30].

The lesion analyses demonstrate the importance of various components. The meta seq2seq learner fails to solve the task without the guidance of the support loss, achieving only 5.43% correct (SD = 7.6). These runs typically learn the consistent, static meanings such as "twice", "thrice", "around right" and "after", but fail to learn the dynamic primitives which require using memory. The meta seq2seq learner also fails when the decoder attention is removed (10.32% correct; SD = 6.4), suggesting that a single $m$ dimensional embedding is not sufficient to relate a query to the support items.

## 4.4 Experiment: Adding a new primitive through augmentation meta-training

Experiment 4.3 demonstrates that the meta seq2seq approach can *learn how to learn* the meaning of a primitive and use it compositionally. However, only a small set of four input primitives and four meanings was considered; it is unclear whether meta seq2seq learning works in more complex compositional domains. In this experiment, meta seq2seq is evaluated on a much larger domain produced by augmenting the meta-training with 20 additional input and action primitives. This more challenging task requires that the networks handle a much larger set of possible meanings. The architecture and training procedures are identical to those used in Experiment 4.3 except where noted.

**Seq2seq learning.** To equate learning environment across approaches, standard seq2seq models use a training phase that is substantially expanded from that in Experiment 4.3. During training, the input primitives include the original four ('jump','run', 'walk', 'look') as well as 20 new symbols ('Primitive1,' ..., 'Primitive20'). The output meanings include the original four ('JUMP','RUN','WALK','LOOK') as well as 20 new actions ('Action1,' ..., 'Action20'). In the seq2seq training (but notably, not in meta seq2seq training), 'Primitive1' always corresponds to 'Action1,' 'Primitive2' corresponds to 'Action2,' and so on. A training batch uses the original SCAN templates with primitives sampled from the augmented set rather than the original set; for instance, a training instruction may be "look around right and Primitive20 twice." During training the "jump" primitive is only presented in isolation, and it is included in every batch to ensure the network learns it properly. Compared to Experiment 4.3, the augmented SCAN domain provides substantially more evidence for compositionality and productivity.

**Meta seq2seq learning.** Meta seq2seq models are trained similarly to Experiment 4.3 with an augmented primitive set. During meta-training, episodes are generated by randomly sampling a set of four primitive instructions (from the set of 24) and their corresponding meanings (from the set of 24). For instance, an example training episode could use the four instruction primitives 'Primitive16', 'run', 'Primitive2', and 'Primitive12' mapped respectively to actions 'Action3', 'Action20', 'JUMP', and 'Action11'. Although Experiment 4.3 has only 23 possible assignments, this experiment has orders-

of-magnitude more possible assignments than training episodes, ensuring meta-training only provides a small subset. Moreover, the models are evaluated using a stricter criterion for generalization: the primitive "jump" is never assigned to the proper action "JUMP" during meta-training.

The test phase is analogous to the previous experiment. Models are evaluated by loading all of the isolated primitives ('jump','run', 'walk', 'look') paired with their original meanings ('JUMP','RUN','WALK','LOOK') into memory as support items. No other items are included in memory. To evaluate test accuracy, models make predictions on the original SCAN test instructions consisting of all composed instructions using "jump."

**Results.** The results are summarized in Table 2. The meta seq2seq learner succeeds at acquiring "jump" and using it correctly, achieving 98.71% correct (SD = 1.49) on the test instructions. The slight decline in performance compared to Experiment 4.3 is not statistically significant with five runs. The standard seq2seq learner takes advantage of the augmented training to generalize better than when using standard SCAN training (Experiment 4.3 and [16]), achieving 12.26% accuracy (SD = 8.33) on the test instructions (with >99% accuracy during training). The augmented task provides 23 fully compositional primitives during training, compared to the three in the original task. The basic seq2seq model still fails to properly discover and utilize this salient compositionality.

The lesion analyses show that the support loss is not critical in this setting, and the meta seq2seq learner achieves 99.48% correct without it (SD = 0.37). In contrast to Experiment 4.3, using many primitives more strongly guides the network to use the memory, since the network cannot substantially reduce the training loss without it. The decoder attention remains critical in this setting, and the network attains merely 9.29% correct without it (SD = 13.07). Only the full meta seq2seq learner masters both the current and the previous learning settings (Table 2).

### 4.5 Experiment: Combining familiar concepts through meta-training

The next experiment examines combining familiar concepts in new ways.

**Seq2seq learning.** Seq2seq training holds out all instances of "around right" for testing, while training on all other SCAN instructions ("around right" split [21]). Using the symmetry between "left" and "right," the network must extrapolate to "jump around right" from training examples like "jump around left," "jump left," and "jump right."

**Meta seq2seq learning.** Meta-training follows Experiment 4.4. Instead of just two directions "left" and "right", the possibilities also include "Direction1" and "Direction2" (or equivalently, "forward" and "backward"). Meta-training episodes are generated by randomly sampling two directions to be used in the instructions (from "left", "right", "forward", "backward") and their meanings (from "LTURN," "RTURN," "FORWARD","BACKWARD"), permuted to have no systematic correspondence. The primitive "right" is never assigned to the proper meaning during meta-training. Meta-training uses both 20 support and 20 query instructions. During test, models must infer how to perform an action "around right" and use it compositionally in all possible ways, with a support set of just "turn left" and "turn right" mapped to their proper meanings.

**Results.** Meta seq2seq learning is nearly perfect at inferring the meaning of "around right" from its components (99.96% correct; SD = 0.08; Table 3), while standard seq2seq fails catastrophically (0.0% correct) and syntactic attention struggles (28.9%; SD = 34.8) [30].

### 4.6 Experiment: Generalizing to longer instructions through meta-training

The final experiment examines whether the meta seq2seq approach can learn to generalize to longer sequences, even when the test sequences are longer than any experienced during meta-training.

**Seq2seq learning.** The SCAN instructions are divided into training and test sets based on the number of required output actions. Following the SCAN "length" split [16], standard seq2seq models are trained on all instructions that require 22 or fewer actions (∼17,000) and evaluated on all instructions that require longer action sequences (∼4,000 ranging in length from 24-28). During test, the network must execute instructions that require ≥24 actions such as "jump around right twice and look opposite right thrice," where both sub-instructions have been trained but the conjunction is novel.

**Meta seq2seq learning.** Meta-training optimizes the network to extrapolate from shorter support instructions to longer query instructions. During test, the model is examined on *even longer* queries

than seeing during meta-training (drawn from the SCAN "length" test set). For meta-training, the original "length" training set is sub-divided into the support pool (all instructions with less than 12 output actions) and a query pool (all instructions with 12 to 22 output actions). In each episode, the network gets 100 support items and must respond to 20 (longer) query items. To encourage use of the external memory, primitive augmentation as in Experiment 4.4 is also applied. During test, the models load 100 support items from the original "length" split training set (lengths 1 to 22 output actions) and responds to queries from the original test set (lengths 24-28).

**Results.** None of the models perform well on longer sequences (Table 3). The meta seq2seq learner achieves 16.64% accuracy (SD = 2.10) while the baseline seq2seq learner achieves 7.71% (SD = 1.90). Syntactic attention also performs poorly at 15.2% (SD = 0.7) [30]. Despite its other compositional successes, meta seq2seq lacks the truly systematic generalization needed to extrapolate to longer sequences.

Table 3: Test accuracy on the SCAN "around right" and "length" tasks.

| Model | around right | length |
|---|---|---|
| meta seq2seq learning | **99.96%** | 16.64% |
| standard seq2seq | 0.0% | 7.71% |
| syntactic attention [30] | 28.9% | 15.2% |

## 5 Discussion

People are skilled compositional learners while standard neural networks are not. After learning how to "dax," people understand how to "dax twice," "dax slowly," or even "dax like there is no tomorrow." These abilities are central to language and thought yet they are conspicuously lacking in modern neural networks [16, 3, 21, 23, 2].

In this paper, I introduced a meta sequence-to-sequence (meta seq2seq) approach for learning to generalize compositionally, exploiting the algebraic structure of a domain to help understand novel utterances. Unlike standard seq2seq, meta seq2seq learners can abstract away the surface patterns and operate closer to rule space. Rather than attempting to solve "jump around right twice and walk thrice" by comparing surface level patterns with training items, meta seq2seq learns to treat the instruction as a template "$x$ around right twice and $y$ thrice" where $x$ and $y$ are variables. This approach solves several SCAN compositional learning tasks that have eluded standard NLP approaches, although it still does not generalize systematically to longer sequences [16]. In this way, meta seq2seq learning is a step forward in capturing the compositional abilities studied in synthetic learning tasks [18] and motivated in the "to dax" or "to Facebook" thought experiments.

Meta seq2seq learning has implications for understanding how people generalize compositionally. Similarly to meta-training, people learn in dynamic environments, tackling a series of changing learning problems rather than iterating through a static dataset. There is natural pressure to generalize systematically after a single experience with a new verb like "to Facebook," and thus people are incentivized to generalize compositionally in ways that resemble the meta seq2seq loss. Meta learning is a powerful new toolbox for studying learning-to-learn and other elusive cognitive abilities [17, 37], although more work is needed to understand its implications for cognitive science.

The models studied here can learn variables that assign novel meanings to words at test time, using only the network dynamics and the external memory. Although powerful, this is a limited concept of "variable" since it requires familiarity with all of the possible input and output assignments during meta-training. This limitation is shared by nearly all existing neural architectures [33, 11, 31] and shows that the meta seq2seq framework falls short of addressing Marcus's challenge of extrapolating outside the training space [24, 25, 23]. In future work, I intend to explore adding more symbolic machinery to the architecture [29] with the goal of handling genuinely new symbols. Hybrid neuro-symbolic models could also address the challenge of generalizing to longer output sequences, a problem that continues to vex neural networks [16, 3, 30] including meta seq2seq learning.

The meta seq2seq approach could be applied to a wide range of tasks including low resource machine translation [13], graph traversal [11], or "Flash Fill" style program induction [28]. For traditional seq2seq tasks like machine translation, standard seq2seq training could be augmented with hybrid training that alternates between standard training and meta-training to encourage compositional generalization. I am excited about the potential of the meta seq2seq approach both for solving practical problems and for illuminating the foundations of human compositional learning.

## Acknowledgments

PyTorch code is available at `https://github.com/brendenlake/meta_seq2seq`. I am very grateful to Marco Baroni for contributing key ideas to the architecture and experiments. I also thank Kyunghyun Cho, Guy Davidson, Tammy Kwan, Tal Linzen, Gary Marcus, and Maxwell Nye for their helpful comments.

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
