[Supplementary Material]

# A Appendix: Compositional generalization through meta seq2seq learning

**Figure A.1 (Query 1 — decoder output)**

| | lug | zup | lug | wif | dax | zup | <EOS> |
|---|---|---|---|---|---|---|---|
| • | 0.02 | 0.05 | 0 | 0.01 | 0.02 | 0.02 | 0.87 |
| • | 0.02 | 0.89 | 0.07 | 0 | 0 | 0 | 0.02 |
| • | 0 | 0.06 | 0.84 | 0.08 | 0.01 | 0 | 0 |
| • | 0 | 0.01 | 0.09 | 0.77 | 0.13 | 0 | 0 |
| • | 0 | 0 | 0.02 | 0.13 | 0.81 | 0.04 | 0 |
| • | 0.02 | 0 | 0.01 | 0.02 | 0.16 | 0.73 | 0.05 |
| <EOS> | 0.01 | 0 | 0.01 | 0.02 | 0.1 | 0.83 | 0.03 |

**Figure A.1 (Query 2 — query instruction, support items)**

| | wif | lug | zup |
|---|---|---|---|
| lug | 0.35 | 0.29 | 0.36 |
| dax | 0.33 | 0.33 | 0.34 |
| dax | 0.34 | 0.33 | 0.34 |
| wif | 0.12 | 0.44 | 0.44 |
| lug | 0.052 | 0.9 | 0.048 |
| <EOS> | 0.18 | 0.64 | 0.19 |

**Figure A.1 (Query 2 — decoder output)**

| | lug | dax | dax | wif | lug | <EOS> |
|---|---|---|---|---|---|---|
| • | 0.03 | 0.05 | 0.01 | 0.01 | 0.06 | 0.8 |
| • | 0.02 | 0.9 | 0.07 | 0 | 0 | 0.01 |
| • | 0 | 0.07 | 0.86 | 0.06 | 0 | 0 |
| • | 0 | 0.01 | 0.09 | 0.86 | 0.04 | 0 |
| • | 0.01 | 0 | 0.02 | 0.25 | 0.66 | 0 |
| <EOS> | 0.01 | 0 | 0 | 0.02 | 0.59 | 0.02 |

Query 1: lug zup lug wif dax zup
Query 2: lug dax dax wif lug
Support set: wif, lug, zup

Figure A.1: During a test episode of the ME task, the support set (top left) and two queries are shown. The ME inference is that "dax" maps to "blue." The key-value memory attention $A$ for each query is shown in the left matrix, with rows as encoder steps and columns as support items. The decoder attention for each query is shown in the right matrix, with rows as the decoder steps and columns as encoder steps. <EOS> marks the end-of-sequence.

**Mutual exclusivity (ME).** Attention is visualized in Figure A.1 for a test episode in the ME task (Experiment 4.2) with two queries "lug zup lug wif dax zup" and "lug dax dax wif lug." When passing a query symbol-by-symbol through the key-value memory (Figure A.1 left), the network allocates attention to all of the cells that *do not* contain the current query symbol, a counterintuitive but valid encoding strategy. This pattern is reversed in the last step before the end-of-sequence symbol (<EOS>), where more intuitively the input symbol activates the memory cell that contains its corresponding support item. The withheld ME symbol "dax" leads to a broad, uniform pattern of attention spread across the support items, indicating its novelty.

The RNN decoder attention is more straightforward. The diagonal pattern indicates strong alignment between each output symbol from the decoder (color; row) and its corresponding input symbol in the encoder (pseudoword; column). The first decoder step is an exception because the decoder hidden state is initialized with the last context step $C_T$ (Section 3). The attention vectors do not sum to 1 because of padded elements from the batched decoder.

**Key-value memory attention**

| | run | walk | jump | look |
|---|---|---|---|---|
| walk | 0.001 | 0.98 | 0.004 | 0.01 |
| left | 0.003 | 0.97 | 0.009 | 0.018 |
| after | 0.038 | 0.81 | 0.041 | 0.11 |
| run | 0.99 | 0.004 | 0 | 0.005 |
| right | 0.98 | 0.01 | 0 | 0.01 |
| thrice | 0.91 | 0.056 | 0.001 | 0.034 |
| <EOS> | 0.62 | 0.21 | 0.01 | 0.16 |

**Decoder attention**

| | walk | left | after | run | right | thrice | <EOS> |
|---|---|---|---|---|---|---|---|
| I_TURN_RIGHT | 0 | 0 | 0 | 0.14 | 0.28 | 0.2 | 0.38 |
| I_RUN | 0 | 0 | 0 | 0.79 | 0.15 | 0.02 | 0.03 |
| I_TURN_RIGHT | 0 | 0 | 0.01 | 0.11 | 0.33 | 0.44 | 0.1 |
| I_RUN | 0 | 0.01 | 0 | 0.67 | 0.23 | 0.07 | 0.02 |
| I_TURN_RIGHT | 0.02 | 0.03 | 0.03 | 0.11 | 0.28 | 0.44 | 0.1 |
| I_RUN | 0.03 | 0.04 | 0.01 | 0.66 | 0.19 | 0.06 | 0.01 |
| I_TURN_LEFT | 0.35 | 0.36 | 0.11 | 0.05 | 0.06 | 0.05 | 0.02 |
| I_WALK | 0.73 | 0.22 | 0.02 | 0 | 0 | 0.01 | 0.01 |
| <EOS> | 0.29 | 0.35 | 0.27 | 0.01 | 0.01 | 0.05 | 0.03 |

Figure A.2: Attention in meta seq2seq learning on the SCAN task. During test, the network is evaluated on the query "walk left after run right thrice." <EOS> marks the end-of-sequence.

**SCAN.** Attention is visualized in Figure A.2 for a test episode in the "add jump" task (Experiment 4.4). The key-value memory attention provides a lookup mechanism for retrieving the response for each input primitive, including "walk" and "run" (Figure A.2 left). The decoder attention also provides an intuitive alignment, attending to "run", "right," and "thrice" in alternation while executing "run right thrice" (Figure A.2 right).