[Reviews · NeurIPS 2019]

Reviewer 1



This work builds on a long debate in cognitive science and AI on systematic and compositional generalization abilities. While previously the debate had to do with appropriate models of cognition, recently the debate has turned towards imbuing machine learning models with the systematic and compositional capacities that humans tend to exhibit. The experiments are based on a previously introduced dataset designed to measure compositional generalization (SCAN). The authors demonstrate substantial improvements over previous results, and they attribute this to a new memory-based architecture and training protocol called meta-seq2seq, which embeds the conventional seq2seq training setup in a meta-learning framework. The overall effect is to “abstract away the surface patterns and operate closer to rule space.”. I believe that the work clearly outlines the methods and model details, and there is no reason to think that anything reported may be technically inaccurate. All-in-all I believe the results are quite a nice demonstration of how learning-based models can exhibit some of the contentious skills so often argued about in the literature. Perhaps someone steeped in meta-learning may think the results are to be expected, but it is nonetheless a novel demonstration of the use of meta-learning for instilling machine learning models with the capacity for systematic and compositional generalization. My primary worry has to do with the framing of the work and some of the phrasing contained within. At a high level, it is motivated and pitched as an addition to a story with the structure “Humans can do X, but machine learning models cannot”. This is evident from the very first sentence of the abstract: “People can generalize in systematic ways from just one or a few examples...In contrast, powerful sequence-to-sequence (seq2seq) neural networks can fail such tests of compositionality“ Though this phrasing isn’t exactly wrong on its surface (indeed, I used similar phrasing in the first paragraph of this review), it nonetheless fails to capture or communicate the nuance needed to properly engage with the underlying debate. Given knowledge of the history of the debate, it is much too easy to call to mind an implicit assumption common in these arguments: humans have a powerful ability that nearly comes for free because of some innate “capacity for compositionality” (this is sometimes even made explicit, such as in the sentence “People can learn new words and use them immediately in a rich variety of ways, thanks to their skills in compositional learning.”). Arguments of this type then go on to suggest that neural networks (in the case of this paper, “standard” neural networks, though I’m not sure what makes one standard or not) fail to match humans in this regard, and hence are deficient in some way. I think we need to be careful with these types of arguments because it is actually not quite clear (a) how good this skill actually is in humans, (b) how much the skill has to be cultivated and honed through long, arduous learning, and (c) whether neural networks as a class of models are indeed deficient or whether we are simply deficient practitioners unable to exploit their full potential. If we are convinced that humans actually need to *learn* this capacity, then it stops being a mystery why neural networks that are not tasked with learning this capacity fail to exhibit it. Indeed, as this paper shows, once a neural network is tasked with learning this capacity it can indeed exhibit it, just like humans! To sum up this previous point and to provide actionable criticism, the authors are encouraged to better contextualize the comparison to human abilities. If they believe that humans are not learning this ability, then they should state so clearly. If they believe that humans are indeed learning this ability, again they should state so clearly, and subsequently explain why previous learning-based models have failed to learn similarly. (Here, I suspect the story isn’t so “flashy”, but nonetheless needs to be told: humans have an immense amount of training experience, much of it as a “meta-level”, and bring in tons of background knowledge knowledge and abilities, whereas neural networks must start “from zero”). If the nuances aren’t appropriately fleshed out, then the argument unfortunately reads as a strawman that the subsequent work sets aflame. Similarly, in the results and discussion the authors are encouraged to write a bit more to contextualize what is happening: through meta-learning, the models are learning *the fact that* primitives that are seen only once/a few times should be treated the same way as others, which is impossible to learn without meta-learning. Again, this is also a potential explanation for how and why humans behave this way; they have a ton of experience learning such facts. The implications of these results should be looped back into the claims made from previous work, such as the paper that introduced the SCAN dataset. Finally, I think that in their argumentation the authors need to keep in mind that they are demonstrating that when a neural network is given an opportunity to learn X, then it will most likely learn X. This is a bit obvious, but too many within this debate lose sight of this fact while trying to argue that neural networks as a class of model are inherently deficient. The results presented here speak directly against this notion and deserve to be emphasized.

Reviewer 2



The paper shows that systematic generalization in seq2seq transduction can be achieved by training a memory-equipped model to rapidly bind symbols to their meanings and use these bindings in the context of a new query. The method is framed as meta-learning, although it is important to note that “learning” in this case essentially just means “loading into memory”. The experiments are conducted on adapted versions of the SCAN and mutual exclusivity tasks from recent literature. The paper is clearly written and the proposed method is easy to understand. The experiments seem to be well executed. My main concern is the significance of the results. As correctly mentioned in the paper in lines 83-87, application of meta-learning has required a massive data augmentation compared to the original SCAN task. After this augmentation is performed, it is rather unsurprising that the resulting model generalizes systematically, as it is forced to treat primitives like “jump”, “walk”, “run” as mere pointers in the memory. Performing such a data augmentation requires a substantial amount of background knowledge, limiting the generality of the proposed method. In particular, one needs to know beforehand which positions in input sequence correspond to which positions in output sequences. It is nontrivial how this can be known for more realistic data, and this question needs to be discussed (like e.g. in “Good-Enough Compositional Data Augmentation”, where a heuristic algorithm for this purpose is proposed). It is furthermore unclear how an appropriate support set would be fetched for a given query in the real world case. In line with the above criticism, I find it rather unsatisfying that results only on toy synthetic tasks are reported. Such tasks are great to highlight weaknesses of existing methods, but they are less appropriate when one proposes a new approach, especially when it has so many underlying assumptions. Lastly, I believe there is a more subtle issue with the proposed method, namely that the design of the data augmention procedure is very strongly informed by the kind of systematicity that is being measured. For example, the specific data augmentation proposed in this paper may or may not help for a hypothetical split in which the word “thrice” is only used in short sentences at training time, but then appears in long sentences during training time. One can construct a meta-learning setup which trains the models to do exactly that: quickly learn (e.g. by loading examples in memory) meanings of words from short sentences and then apply the learned meanings in long ones. But there is an infinite number of ways in which test data can be systematically different from training, and arguably, all of them can’t be anticipated. To conclude, I think the current version of the paper needs improvement. In particular, it should explain and empirically prove the significance and the generality of the proposed method. A few related papers: - https://arxiv.org/abs/1904.09708 proposes a simple approach to get a high performance on SCAN and should be cited (even though it seems like a concurrent work). - https://arxiv.org/abs/1703.03129 is relevant in that it uses a memory is used to quickly learn meanings of words UPD. I have read the rebuttal and other reviews. I still think that the fact that the network exhibits the desired behavior is rather unsurpising after 23 possible meaning permutations are used to effectively augment the data. The authors have suggested that they are working on applying the proposed method in other settings, namely few-shot language modelling and Flash-fill program induction, and that in those settings a “compositionally-informed episode-generator” is not required. While this sounds quite interesting, these experiments are not a part of the paper. I find the results of the paper insufficient to believe without evidence that the proposed method will work when a compositionality-informed episode generator is not used. More broadly, I think there are 2 ways to look at this paper. If it is viewed as a cognitive science paper, then like R1 correctly mentioned, better positioning would be required. Can the proposed protocol be viewed as a model of child language acquisition? Seems dubious to me, but I am not an expert. Can we conclude that essentially neural networks can learn the rules like humans do but we just have not been giving them the right data? In my view the training episodes in the paper may contain more evidence for compositionality than any real world data stream would. This needs to be discussed in more depth, and the paper currently doesn’t do so. Alternatively, the paper can be evaluated as a machine learning work, that allegedly addresses an important shortcoming of seq2seq models. In such case, I think the paper should have shown that the proposed method can be useful in the absence of or with a less task-specific “compositionally-informed episode-generator”.

Reviewer 3



** SUMMARY * Goal: A NN learning approach that produces a high degree of compositional generalization (in a specific type of setting). * Setting: In seq2seq tasks, if a model has learned that the input 'x twice' should produce the output 'Y Y', for a given set of input/output token pairs x/Y, then given a single example showing that a new token 'b' should produce output 'Q', the model should without training map 'b twice' to 'Q Q'. More generally, a single model must deal with multiple 'episodes' of seq2seq training/testing. In a single episode, the mapping from input to output tokens is fixed. Across episodes, certain input tokens (which I'll call "lexical") map to different output tokens, while other input tokens (which I'll call "functional") map consistently to operations over output tokens, in all episodes. (In the example above, 'b' is lexical, 'twice' is functional.) Within an episode, example I/O pairs show that particular episode's mapping of lexical input tokens to their corresponding output tokens. * Model innovation: Provide a memory that, in each episode, can store given episode-specific input/output pairs (called the episode's "support") which can be used to determine the episode-specific mapping of lexical input tokens (which I consider "declarative knowledge"). (Key-value-query) attention over this memory can retrieve the episode-specific output correspondents of lexical input tokens, while fixed weights learned across multiple training episodes encode (as what I consider "procedural knowledge") the meaning of functional tokens (like 'twice'). * Complexity: The innovation does not entail new complexity issues. * Tested models: In a given episode, for each of a set of I/O "support" examples, the final state of a non-episode-specific bi-LSTM encodes the example's input sequence as a vector which serves as a key to a memory slot, the value of which is the final state vector of a second non-episode-specific bi-LSTM encoding the example's output sequence [Fig 1]. The actual test input string (as opposed to the support inputs) is processed by a third LSTM. Each input token generates a query vector which retrieves information from the memory containing the support examples. The retrieved information (along, I believe, with the query vector) is accumulated across the whole input string. The resulting matrix is attended to by a fourth LSTM which generates the output tokens. * Tasks: - Expts 1-3: Versions of the SCAN compositionality task, mapping an input string (e.g., 'jump twice') describing an action sequence to the described sequence of primitive actions (e.g. 'JUMP JUMP'). The input strings combine lexical tokens (e.g., 'jump') referring to primitive actions (like 'JUMP') with a set of functional modifiers (including 'twice', 'around right', 'and', 'after'). (Expts 1, 2 have 4, 24 types of lexical tokens, respectively.) Across episodes, the meaning of the functional tokens remains constant (e.g., 'twice' always means twice) but the meaning of the lexical tokens is permuted to a unique mapping between input and output tokens (e.g,'jump' denotes primitive action 'RUN' in episode 1, it denotes 'LOOK' in episode 2, etc.). One mapping is withheld from training and used as the test episode: 'jump' -> 'JUMP', 'run' -> 'RUN'", etc. In Expt 3, withheld from training are *all* permutations that map the specific token 'jump' to 'JUMP' (including permutations that map some other lexical token 'x' to 'Y' =! 'X'); tested are all input sequences containing 'jump'. In each training episode, a support set of 20 randomly-chosen input sequences and their corresponding output sequences is provided from which the episode-specific meaning of lexical tokens can be deduced. In the single test episode, the support contains 4 I/O pairs, each consisting of a single input token and its corresponding output token. - Expt 4: 4 input tokens are mapped to 4 output tokens, differently across episodes; the support set in each episode consists of 3 I/O pairs each containing a single input and single output token: the fourth pairing is not in the support, and must be deduced from the Mutual Exclusivity generalization, exhibited during training, that each output corresponds to only a single input token, so the output token missing from the support must correspond to the input token missing from the support. * Results: In Expts 1-3 [Table 2], over all complex instructions containing 'jump', at test a standard seq2seq model achieves 0.03% (whole-output-string) accuracy while the meta seq2seq model performs at 99% on Expt 3 and 100% on Expts 1-2. A strong Mutual Exclusivity bias emerges in Expt 4, with an accuracy of 96%, indeed "showing it can reason about the absence of a symbol in addition to the presence of a symbol" [300]. ** REVIEW * The problem of compositional generalization addressed here is very important. * The results are striking. * What is demonstrated is that, in a seq2seq task, if the training data show that the I/O mapping contributed by certain input tokens varies, then a NN can learn to treat those tokens as variables, using a key/value/query memory retrieval system (a.k.a. "attention") to look up the values of those variables. This is true even when the mapping between input and output tokens is only indirectly specified via complex-input-sequence to complex-output-sequence examples. I think this is an important result. * It's not clear that people require the kind of training given here, explicitly being shown evidence that 'jump' could mean entirely different things in different contexts, while 'twice' always means the same thing, before they can use 'jump' compositionally with 'twice'. I'm not sure what to make of that, though. * It would be good to consider this work in the light of work on program synthesis, program execution, and mathematics problem tasks. Answering math questions or producing the output of computer programs also involve binding values to variables, but there the assignment of values to variables is explicit in the input to the network. In the present paper, the problem is more like example-based program synthesis, where a sample of I/O pairs of an unknown program' (like 'jump twice' -> 'JUMP JUMP') is the model's input, and the model needs to determine the implicit binding of values to variables in order to produce the output of the program for novel inputs. Here, however, the program is hardwired into the network connections, and not a symbolic output of the model. * The distinction among input tokens I'm calling lexical vs. functional should be made more explicitly in the paper. I find the term "primitive instructions" confusing because the instructions consist of sequences of tokens which are the primitives of instructions, and these include both what I'm calling lexical and functional tokens. * The term "meta-learning" is confusing, to the extent that that means "learning how to learn", because there is not learning in the testing episode. During training, it's just learning how to use a memory during testing to find the current value of a variable: that is, learning how to *process*, not learning how to *learn*. * It would be good to insert "implicit" into "Our approach ... can learn to apply [implicit] rules to variables." [59] as this is not about explicit rule interpretation. * The description of the precise activation flow in the model is unclear (hence my hedging when attempting to summarize it): a proper, complete set of equations should be provided in Sec. 3. My problem may be that I'm not familiar with "Luong-style attention" and don't have time to read the cited paper. * "During training, models are exposed to the “jump” instruction in only a single training pattern when the instruction “jump” appears in isolation" [164] is confusing given the later "Episodes consist of 20 support and 20 query instructions sampled uniformly without replacement from the complete set of SCAN patterns" [179] (It's also unclear what a "SCAN pattern" is, exactly.) [ADDED NOTE: I read the authors' feedback, the other reviews, and reviewer discussion several times each, and pondered for a considerable time. However, in the end, I see no reason to change my score or review, except to acknowledge here that the proposed revisions in the authors' feedback concerning a few of the points in my review would all improve the paper: thank you. There are a number of points in my review that were not addressed, but this is inevitable given a single page to reply to three reviews. If more of the points I raised in the review can be remedied in a revision, so much the better.]

[Author Response · NeurIPS 2019]

We thank each of the referees for taking the time to carefully read and comment on our work. We are pleased to see our paper was generally well-received, and that the problem "addressed here is very important" (**R3**) and that it "addresses some issues in a highly contentious, and long-standing debate in the cognitive science and AI literature" (**R1**). We are also happy that our findings are compelling: "the results are striking" (**R3**) and provide "a novel demonstration of... instilling machine learning models with the capacity for systematic and compositional generalization" (**R1**). We, too, are excited about meta seq2seq learning and its ability to acquire compositional skills.

The reviewers raise the following primary issues. **R1** discusses the framing of the paper and asks us to "provide more nuance and context regarding the [debate] on compositional and systematic generalization." **R2** asks us to further "explain how the proposed method can be applied to natural data." Finally, **R3** asks clarification questions and how meta-training can be interpreted in terms of human learning. We see these points as readily addressable.

**Compositionality in humans and machines.** Thanks **R1** for the thoughtful suggestions on framing, which we will happily incorporate in our revisions. We began the paper by contrasting the compositional skills of humans and machines. **R1** asks "How good this skill actually is in humans?" – indeed, recent work shows it is quite good! [1] In our remarks on compositional learning, we did not intend to take a stance on the *origin of these human abilities*. Certainly we do not want to suggest that "humans have a powerful ability that nearly comes for free because of some innate 'capacity for compositionality'", as **R1** asks us to clarify. People and models differ substantially in their experience and background knowledge, and we will make this absolutely clear in our revisions.

**R1** also asks whether we believe compositional skills are learned or innate (nature vs. nurture). Relatedly, **R3** wonders whether people require meta-training to generalize compositionally. We are very glad that our paper stimulates these fascinating questions. We can't resolve them here, as they are empirical questions, but there is compelling evidence that learning (even meta-learning) plays a role. We will add this discussion to the paper: First, infants have limited compositional skills [4] which improve with age [3]. Second, some language-related inductive biases are either learned or develop [6, 2], and compositionality could have similar origins. Third, we provide *a novel demonstration of how agents could learn compositional skills through experience*, which is a key contribution of our paper.

**R1** and **R3** ask whether people could be doing meta-learning, and indeed this as a real possibility. People have experience at the "meta-level," although it is quite different than meaning permutation we used for SCAN. As we see it, there is a "natural pressure to generalize systematically after a single experience with a new verb like 'to Facebook,'" (pg. 8): a child hears one or a few uses of a novel word (the support), and she must be able to use the new word properly in new sentences (the queries). As with our meta-training, people are incentivized to generalize compositionally from a brief experience with a new word. We are currently applying our approach to few-shot word learning (language modeling) problems of this flavor, which relates to the comments of **R2**.

**Naturalistic data.** First, we respectfully disagree with **R2**'s assessment that the net learns "jump" and "walk" as mere pointers in memory; it's substantially more complex since the support set only indirectly specifies these mappings through multi-word commands (not isolated primitives). We will make revisions to say this more clearly. Further addressing **R2**'s comments, meta seq2seq learning is a very general framework with applications to other domains. To solve SCAN, meta seq2seq requires a compositionally-informed episode-generator, but this is a property of SCAN rather than an inherent property of meta seq2seq. The few-shot language modeling task, described in the paragraph above, does not require any special knowledge to generate episodes besides a set of sentences that all use the same new word. As another example, we are currently working on an application to "Flash Fill"-style program induction problems. During meta-training, one episode could show several dates formatted like "05/05/1987" mapped to the format "May 5, 1987", and another episode might shows dates like "1987/05/05" mapped to "month:may year:1987". Meta seq2seq would acquire compositional skills that support few-shot learning of new mappings/programs at test time, such as "05.05.98" to "fifth of May, 1987." Similarly, practical applications such as learning transformations of names, numbers, locations, etc. require no special knowledge for episode generation and require no ground-truth program supervision.

**Other revisions.** We will cite concurrent work from [5] and add a direct comparison in the paper (**R2**). On the "add jump" task, our method achieves a mean accuracy of 98.7% (Exp 3) while [5] achieves 78.4%. We also ran new experiments on the SCAN "around right" split, and our method achieves 99.96% and the other 28.9% [5].

Thanks **R3** for pointing out a potential confusion: line 164 refers to seq2seq training and line 179 refers to meta seq2seq training. We will reorganize the methods into two separate sections for "seq2seq training" vs. "meta seq2seq training." Also the equations for Luong-style attention are on lines 138-140, which we will unpack better in our revisions.

We thank you for considering our work in your further discussions.

[1] B. M. Lake, T. Linzen, and M. Baroni. Human few-shot learning of compositional instructions. In *Proceedings of the 41st Annual Conference of the Cognitive Science Society*, 2019.
[2] M. Lewis, V. Cristiano, B. M. Lake, T. Kwan, and M. C. Frank. The role of developmental change and linguistic experience in the mutual exclusivity effect. *PsyArXiv preprint*, 2019.
[3] S. Piantadosi and R. Aslin. Compositional reasoning in early childhood. *PLoS ONE*, 11(9):1–12, 2016.
[4] S. T. Piantadosi, H. Palmeri, and R. Aslin. Limits on Composition of Conceptual Operations in 9-Month-Olds. *Infancy*, 23(3):310–324, 2018.
[5] J. Russin, J. Jo, R. C. O'Reilly, and Y. Bengio. Compositional generalization in a deep seq2seq model by separating syntax and semantics. *arXiv preprint*, 2019.
[6] L. B. Smith, S. S. Jones, B. Landau, L. Gershkoff-Stowe, and L. Samuelson. Object name learning provides on-the-job training for attention. *Psychological Science*, 13(1):13–19, 2002.


[Meta-Review · NeurIPS 2019]

The reviewers agree that this paper is sound and of potential interest to some audiences. They disagree about whether the proposed solution (meta learning based on a large augmented data set) makes the result boring or interesting. Given that the results appear sound and are likely to yield interesting discussions at the conference, as they have among the reviewers, I recommend including it. I hope the authors revise the framing and discussion after taking into account the reviewers' comments, especially R1.